# Boron Doped Diamond for Real-Time Wireless Cutting Temperature Monitoring of Diamond Coated Carbide Tools

**DOI:** 10.3390/ma14237334

**Published:** 2021-11-30

**Authors:** Sérgio Pratas, Eduardo L. Silva, Miguel A. Neto, Cristina M. Fernandes, António J. S. Fernandes, Daniel Figueiredo, Rui F. Silva

**Affiliations:** 1Department of Materials and Ceramics Engineering, CICECO—Aveiro Institute of Materials, University of Aveiro, 3810-193 Aveiro, Portugal; sergiopratas@ua.pt (S.P.); elsilva@ua.pt (E.L.S.); mangelo@ua.pt (M.A.N.); 2Palbit S.A., P.O. Box 4, Branca, 3854-908 Albergaria-a-Velha, Portugal; cfernandes@palbit.pt (C.M.F.); dfigueiredo@palbit.pt (D.F.); 3Department of Physics, I3N, University of Aveiro, 3810-193 Aveiro, Portugal; toze2@ua.pt

**Keywords:** CVD diamond, boron-doped diamond, thermal sensors, carbide tools

## Abstract

Among the unique opportunities and developments that are currently being triggered by the fourth industrial revolution, developments in cutting tools have been following the trend of an ever more holistic control of manufacturing processes. Sustainable manufacturing is at the forefront of tools development, encompassing environmental, economic, and technological goals. The integrated use of sensors, data processing, and smart algorithms for fast optimization or real time adjustment of cutting processes can lead to a significant impact on productivity and energy uptake, as well as less usage of cutting fluids. Diamond is the material of choice for machining of non-ferrous alloys, composites, and ultrahard materials. While the extreme hardness, thermal conductivity, and wear resistance of CVD diamond coatings are well-known, these also exhibit highly auspicious sensing properties through doping with boron and other elements. The present study focuses on the thermal response of boron-doped diamond (BDD) coatings. BDD coatings have been shown to have a negative temperature coefficient (NTC). Several approaches have been adopted for monitoring cutting temperature, including thin film thermocouples and infrared thermography. Although these are good solutions, they can be costly and become impractical for certain finishing cutting operations, tool geometries such as rotary tools, as well as during material removal in intricate spaces. In the scope of this study, diamond/WC-Co substrates were coated with BDD by hot filament chemical vapor deposition (HFCVD). Scanning electron microscopy, Raman spectroscopy, and the van der Pauw method were used for morphological, structural, and electrical characterization, respectively. The thermal response of the thin diamond thermistors was characterized in the temperature interval of 20–400 °C. Compared to state-of-the-art temperature monitoring solutions, this is a one-step approach that improves the wear properties and heat dissipation of carbide tools while providing real-time and in-situ temperature monitoring.

## 1. Introduction

Nowadays, three main factors drive the need to increase productivity in metal-machining companies: (i) the development of special alloys with tight machining requirements, (ii) the increase in global market competition fueled by cheap labor costs in emerging economies, and (iii) the rise of additive manufacturing. According to Astakhov and Outeiro [1], machining time is now the bottleneck of modern manufacturing. Considering that in an automotive shop tool costs represent only 3% of the total manufacturing cost, there is a great potential to increase productivity by using advanced tools that allow higher cutting speeds and/or feed rates [1]. Simultaneously, an increase in productivity can be achieved by optimization of the cutting cycle, with the objectives of improving tool life and the minimization of downtime. To achieve this, it is paramount that variables, such as cutting force and cutting temperature, are monitored as accurately as possible. Heat generation during machining occurs in three main zones [2]. The primary shear zone generates maximum heating due to plastic deformation of the metal. In the secondary shear zone, heat originates from friction between the tool and the moving chip. At the tertiary deformation zone, heat is mainly produced due to the work required to overcome friction between the flank face and the machined surface [2].

Several approaches to in situ temperature monitoring have been attempted, such as the use of infrared (IR) thermography and thermocouples. Rizal and co-workers developed a multisensory device for simultaneous measuring of torque, vibration, temperature, and cutting force [3]. The temperature sensor consisted of an embedded type-K thermocouple placed below the cutting insert. Data transmission was conducted via a wireless telemetry system based on the inductive principle. The authors successfully demonstrated temperature monitoring for milling of AISI P20 tool steel with a sensitivity of 0.173 μV/°C [3]. Campidelli et al. investigated temperature monitoring during the milling of AISI D2 steel with TiN/TiCN coated carbide inserts [4]. A type-K thermocouple embedded within an indexable insert, near the cutting edge, was used to record the temperature. Sensitivity to the variation of cutting parameters was demonstrated, with higher temperature corresponding to higher values of cutting speed and feed rate. At a cutting speed of 180 m/min, the embedded sensor recorded temperatures of 152, 163, and 180 °C at feed rates of 0.1, 0.2, and 0.3 mm/rev, respectively [4].

Yang and collaborators presented a different approach based on temperature monitoring with an infrared pyrometer [5]. This non-contact method was employed by drilling holes in the workpiece with visibility to the area being machined. By monitoring the temperature of the rake and flank faces through the holes, at a cutting speed of 130 m/min, the authors found that the average temperature of the rake face is 47.5 °C higher than flank face [5].

Although temperature monitoring via the Seebeck effect or IR sensors does provide good approximations of the real surface temperature, both involve considerably elaborate setups. Temperature monitoring with thermographic cameras is very sensitive to the emissivity of materials, which depend on multiple factors like surface roughness, measurement angle and even the temperature of the surface being measured [6]. Hence, an accurate measurement can only be attained if the emissivity of the object being measured is well known throughout each specific range of varying machining conditions [6].

In the case of thermocouples, temperature measurement implies the use of specifically designed tools that can physically accommodate the thermocouple wiring. Such temperature acquisition design increases the cost of the tool, affecting its mechanical properties. It also does not provide an accurate thermal reading, since the hot junction cannot be positioned directly at the cutting interface, where the maximum temperature occurs, along with an extremely sharp gradient. Consequently, the temperature decay between the cutting interface and the hot junction reading is very high, and only average temperature readings can be measured [2]. Thus, while providing a reasonable approximation, thermocouples are not suited for in situ temperature measurement during machining operations. Thermistors are temperature sensors characterized by temperature-dependent resistance variation. These present a fast response time as well as higher sensitivity and accuracy than thermocouples and platinum resistive temperature detectors [7]. Boron doped diamond (BDD) thin films grown by chemical vapour deposition (CVD) exhibit sensitivity to temperature characterized by a negative temperature coefficient (NTC), i.e., when the temperature increases the electrical resistance of the thermistor decreases. Several studies have demonstrated the performance of boron doped CVD diamond based thermistors, which besides high thermal sensitivity also exhibit high chemical inertness, thermal conductivity, and mechanical robustness, making them ideal candidates for applications in harsh environments [8,9,10,11,12]. In this work, BDD thermistors were used for real-time temperature monitoring of end milling operations, with wireless signal communication based on LED-to-LED communication, or “Li-Fi” (light fidelity). A comparison between infrared thermography and the thermistor outputs demonstrated that the BDD thin film thermistors exhibit a fast response and are suitable for real time measurements in static or rotating configurations.

## 2. Materials and Methods

### 2.1. Deposition of Undoped Diamond on Carbide Tool

A micro ball end mill (Palbit, S.A., Albergaria-a-Velha, Portugal) composed of cobalt cemented tungsten carbide (WC 0.5 μm), with 7 wt.% Co and shaft diameter of 2.5 mm was preconditioned for cobalt etching and surface roughening. The WC grains were attacked with Murakami reagent (10 g KOH + 10 g K_3_Fe(CN)_6_ + 100 mL water) during 15 min for surface roughening. Subsequently the tools were immersed in H_2_SO_4_ 1:14 H_2_O_2_ during 3 s, for cobalt etching. The pre-treated samples were seeded with diamond powder and coated with multilayer diamond films in a hot filament assisted CVD (HFCVD) reactor. Tungsten filaments (⌀ = 150 μm) were used for gas activation and the reaction chamber exhibited a cylindrical geometry, with water cooling. Prior to the growth stage, the filaments were carburized during 30 min at 2300 °C and a CH_4_/H_2_ ratio of 0.05. Subsequently, growth conditions were employed, and the films were composed of nanocrystalline diamond (NCD) and submicrocrystalline diamond (SMCD) according to the parameters in Table 1.

### 2.2. Thermistor Fabrication

The deposition of the boron doped diamond film for temperature monitoring as an NTC thermistor was also done by HFCVD, according to the growth conditions in Table 1. The BDD film was grown on top of a silicon nitride (Si_3_N_4_) insulating substrate. The doping process was carried out by using Argon as carrier gas for the boron dopant, by bubbling it through a B_2_O_3_/ethanol solution with B/C ratio of 10,000 ppm. The electrical contacts were also deposited by HFCVD. These must ensure ohmic behavior while displaying good mechanical robustness. A method for growing ohmic contacts was previously developed [8] and is replicated in this work. In short, tungsten oxide (WO_2_) was vaporized from the tungsten hot filaments of the HFCVD system at a temperature of 1800 °C under a primary vacuum of 0.08 kPa. Subsequently, hydrogen and methane were introduced in the chamber, leading to the reduction of WO_2_ and formation of ohmic tungsten carbide WC contacts. A sketch of the experimental setup and growth conditions for undoped and doped CVD diamond is presented in Figure 1.

### 2.3. Microstructural and Electrical Characterization

Both the undoped and boron doped diamond coatings from the tool and thermistor, respectively, were characterized by SEM (Hitachi SU-70, Tokyo, Japan) and Raman spectroscopy (Jobin-Yvon LabRaman HR with 441.6 nm laser line). The electrical resistance was measured as a function of temperature, using a Keithley 2410 electrometer (Keithley Instruments, Solon, OH, USA). The specimens were heated at a constant rate of 10 °C/min, from 50 to 400 °C.

### 2.4. Machining and Temperature Monitoring

The diamond coated micro end mills were tested for end milling of an Inconel 718 workpiece (Figure 1), using a CNC miller (CNC Makino IQ300). The following cutting conditions were used: Cutting speed (V_c_) = 5.48 m/min; Spindle speed (n) = 30,000 rpm; Axial depth of cut (a_p_) = 0.03 mm; Radial depth of cut (a_e_) = 0.03 mm; Feed rate (f_z_) = 6 µm/tooth.

During the machining process, the temperature was monitored simultaneously by the developed diamond-based thermistor and by an IR thermographic camera (FLIR Systems, Wilsonville, OR, USA.). The maximum temperature plot registered by the IR camera was then correlated with the voltage output registered by the thermistor over the machining time. The resistance variation measured by the diamond thermistor in response to the infrared radiation generated during the machining process, was transmitted via the so-called Li-Fi communication. A high brightness LED coupled to the tool holder by a 3D printed support, was used to transmit the resistance change measured by the diamond thermistor. A photodiode placed in a fixed support in close proximity to the spinning emitter LED was used as light sensor, wirelessly transforming the resistance variation of the thermistor into a voltage reading.

## 3. Results and Discussion

### 3.1. CVD Diamond Morphology

The undoped CVD diamond films presented a multilayer architecture (9 layers) by intercalation of two different morphologies, nanocrystalline diamond (NCD) and submicrocrystalline diamond (SMCD) (Figure 2a). The reason underlying the choice for this layered structure is related to the intrinsic properties of diamond and the requirements of the machining process. Since diamond is the hardest known material, it is also fragile and susceptible to catastrophic failure when subjected to dynamic loads while cutting, especially when grown on materials with high dissimilarity in thermal expansion. While the range of possible diamond coating morphologies includes ultrananocrystalline diamond (UNCD), NCD, SMCD and microcrystalline diamond (MCD), the latter morphology exhibits the largest grain size and roughness, and usually the highest hardness as well [13]. However, high roughness also comes with a higher friction coefficient, cutting forces, and stress concentration on first contact with the workpiece, which can lead to catastrophic delamination of the diamond coating and loss of anti-wear protectiveness for the tool [14,15,16]. Therefore, one possible strategy to improve the performance of both coating and tools is to increase the density of grain boundaries of the coating, while still accounting for its desired function. Hence, instead of MCD, a combination of SMCD and NCD can be a better alternative leading to a coating with a combination of low roughness, high hardness, and high adhesion. The combination of a high grain boundary density and layered structure enables a greater capability for stress relaxation in diamond multilayer coatings. Furthermore, the resistance to crack propagation is higher, leading to progressive layer-by-layer wear instead of coating delamination [17]. For the thermistor fabrication, an MCD monolayer coating (average crystallite size = 0.81 μm) was used because higher doping efficiency can be attained, and therefore the resistance of the film is easier to regulate (Figure 2b) [18].

The undoped diamond coated micro end mill shows the described intercalated layered structure with a top layer of NCD (Figure 2a). This was confirmed by Raman spectroscopy where a typical NCD spectrum was recorded, i.e., low ratio between the intensity of the diamond peak and that of the G band (Figure 2c). Conversely, the spectrum for the BDD coating from the thermistor shows a much more pronounced diamond peak (Figure 2d), in line with its larger crystallite size, as corroborated by SEM.

### 3.2. Boron Doped Diamond Electrical Resistance

The non-linear resistance temperature dependence found in NTC thermistors can be fitted by the so-called beta model (Equation (1)), an Arrhenius type dependence, limited to small temperature variations (most NTC thermistor manufacturers specify *B*-values between the standard 25–100 °C range) [7]. The linearization of this equation allows the determination of the sensitivity of the thermistor, quantified by the β parameter (Equation (2)). An alternative model fitting the dependence of thermistor resistance with temperature is the Steinhart–Hart equation, a third order polynomial that can better fit the variation of resistance over a wider temperature range (Equation (3)) [19,20]. Accordingly, both models were used to fit the temperature dependent resistance of the BDD film (Figure 3).
(1)$$R\left(T\right)=Ae^{\left(\frac{\beta }{T}\right)}\;$$
(2)$$\beta =\frac{\ln\left(\frac{R_{1}}{R_{2}}\right)}{\left(\frac{1}{T_{1}}-\frac{1}{T_{2}}\right)}$$
(3)$$\frac{1}{T}=A+B\ln\left(R\right)+C\ln{\left(R\right)}^{3}$$

Figure 3a depicts the resistance temperature dependence for the produced thermistor, recorded in the 25–400 °C temperature range. The resistance of the thermistor varies from around 600 kΩ at room temperature to approximately 10 kΩ at 400 °C. This resistance-temperature dependence is well within the range of practical application for NTC thermistors, since higher or lower resistance values can have a negative impact on the sensitivity of the device [21]. The curve in Figure 3a was linearized according to Equation (2), where β, which indicates the sensitivity of the thermistor, can be determined by the corresponding slope (a steeper curve corresponds to higher resistance variation per 1 °C). As expected, the curve could not be fitted by a straight line over such an extensive temperature range, but three linear regions could be identified, each corresponding to a distinct β.

The BDD thermistor presents higher sensitivity in the higher temperature range, as shown by the highest β of 3500 K. This is in contrast with the work of other authors, which determined higher β at lower temperatures [8]. Most likely, this behavior is related to a hopping conduction mechanism at low temperatures, which is in agreement with the low doping level of the diamond film (~600 kΩ at 25 °C). This corresponds to a resistivity of ~900 Ω.cm (thickness of BDD film ~15 μm), and a boron concentration in the BDD film in the range of 10^14^ cm^−3^ [22].Considering this behavior, β is lower in the low temperature range because the diamond film resistance varies with T^−1/4^ instead of T^−1^ [10,23,24]. In doped semiconductors, electronic conduction at low temperatures occurs through impurity states. In the case of heavily doped semiconductors, conduction occurs through the formation of an impurity band [25,26]. At low dopant concentrations, impurity states are localized and therefore electrons or holes migrate by hopping from occupied to unoccupied states with the assistance of phonons. This process is heavily dependent on thermal activation, and therefore at low temperatures there is a sharp decrease in electrical conductivity [23,25,26]. Accordingly, the Steinhart–Hart model is a good fit for the experimental data, particularly in the 250–400 °C range, below which there is considerable deviation from the measured data.

### 3.3. Wireless Temperature Monitoring during Milling

With the purpose of testing the applicability of the BDD thermistor for real time temperature monitoring during machining, the diamond coated carbide tool was used for face milling of an Inconel 718 workpiece. The BDD thermistor was placed as close as possible to the shear plane, where most of the heat is generated during the machining process (Figure 4a). A dedicated adapter was made by additive manufacturing with the purpose of housing the thermistor and the high brightness LED emitter circuit. This LED emitter was powered by a 3V battery, and its brightness varied according to the temperature-dependent resistance variation imposed by the BDD thermistor (Figure 4b). Since the thermistor circuit was coupled to the tool holder, it was submitted to the same rotating conditions as the tool. A second adapter was coupled to the body of the CNC miller to provide support to the photodiode and amplifier/modulator circuit (Figure 4). This adapter was fixed, and its function was to receive and process the wireless signal acquired through the brightness variation of the LED emitter. Face milling of the workpiece was performed in a back-and-forth linear motion in the X-Y plane, without increment in the Z direction, and with the tool permanently within the boundaries of the workpiece. During the machining process, simultaneous temperature monitoring was performed by IR thermography and by the diamond-based thermistor (Figure 4a). The IR thermographic camera was placed 30 cm away from the tool, with the emissivity set to 0.95, allowing the calibration of the thermistor in terms of voltage-temperature variation, and response time as well. Correspondence between the thermal imagery and the temperature plot shows that the highest temperatures were registered at the end mill tip when the tool reached inflection positions at both the extremities of the cutting path, and lowest temperatures were registered at mid path position (Figure 5). This was expected since the tool was constantly machining within the boundaries of the workpiece. When the tool reached the inflection points at the extremities of the machining path, the movement ceased for a brief period, leading to a temperature increase at such positions. The difference in maximum registered temperature at the left and right turning points was attributed to slight tilting of the workpiece surface.

The diamond thermistor response closely matched the response time of the thermographic camera with an average delay of approximately 1 s (Figure 5). The IR camera registered an average temperature difference of 20.25 °C between the tool tip at mid path and at the inflection positions. The temperature curve was used for calibration of the thermistor (Figure 5). The diamond-based sensor induced an average voltage variation of 16.5 mV in the photodiode, corresponding to the difference between lowest temperature at mid path and highest temperature at inflection positions of the end mill. This means 0.84 mV correspond to a 1 °C shift in temperature. The response of the thermistor, however, does not exhibit the same regularity as registered by the thermographic camera, as clearly shown by the overlapping of both curves (Figure 5). Among the possible causes for this behavior, a limitation of the wireless communication system was identified as a major constraint. Above a spindle speed of 4000 rpm, signal transmission was disrupted, as the brightness of the LED emitter started to fade. Consequently, since the spindle speed could not be increased, the heat generated during the machining process was not high enough for the thermistor to operate at higher temperatures and β parameter, i.e., with higher sensitivity.

Nevertheless, the feasibility of wireless temperature monitoring through the use of boron doped diamond-based thermistors and a “Li-Fi” wireless communication system was clearly demonstrated. The works of Pel and collaborators demonstrate that the sensitivity and response time of the diamond thermistor can be improved [11,12]. An upcoming work will address the improvement of such parameters through the direct inclusion of a boron doped diamond thermistor layer on the coated tool, envisioning the fabrication of advanced diamond tools in line with the concept of Industry 4.0.

## 4. Conclusions

Boron doped diamond-based thermistors were fabricated by HFCVD and used for temperature monitoring during the face milling of an Inconel 718 workpiece with a diamond coated carbide end mill. The room temperature resistance of the thermistor was approximately 600 kΩ. The linear regression of the characteristic resistance-temperature dependence of the BDD thermistor allowed the determination of three temperature sensitivity regimes translated by the magnitude of the β parameter. β was highest for the temperature range of 250–400 °C, reaching 3500 K.

LED to LED communication (Li-Fi) was successfully used for conversion and transmission of the variation of thermistor resistance with temperature, although limited by a maximum spindle speed of 4000 rpm.

Calibration of the diamond-based thermistor by IR thermography demonstrated that the produced sensor exhibits a fast response, with a delay of approximately 1 s by comparison, and an average variation of 0.84 mV/°C.

## Figures and Tables

**Figure 1 materials-14-07334-f001:**
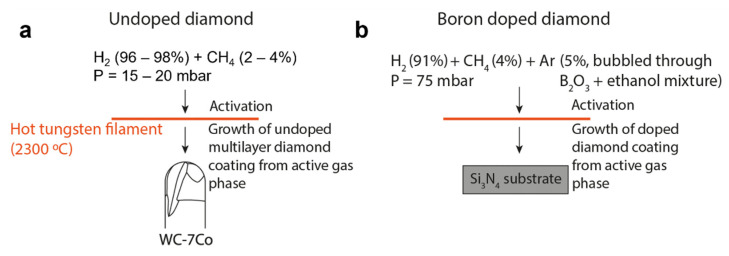
Experimental setup for Hot Filament Chemical Vapor Deposition of (**a**) undoped diamond on carbide end mills and (**b**) boron doped diamond on a silicon nitride substrate for thermistor fabrication.

**Figure 2 materials-14-07334-f002:**
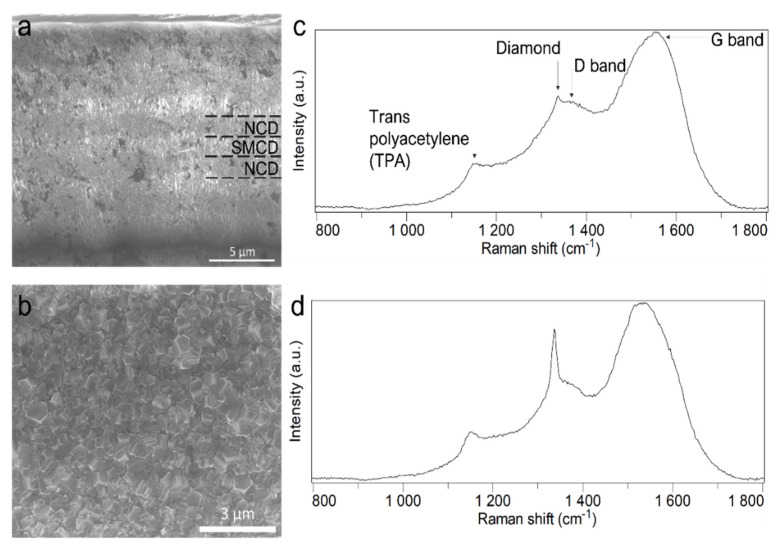
Characterization of the synthesized diamond films by HFCVD. Scanning Electron Microscopy of (**a**) cross section of the undoped diamond multilayer coating on top of the WC-7Co tool and (**b**) surface of the boron doped diamond (BDD) film used for the thermistor. Raman spectroscopy of the (**c**) top layer of the undoped coating, showing a typical NCD spectrum and the (**d**) BDD coating showing a more prominent diamond peak due to the larger crystallite size of MCD.

**Figure 3 materials-14-07334-f003:**
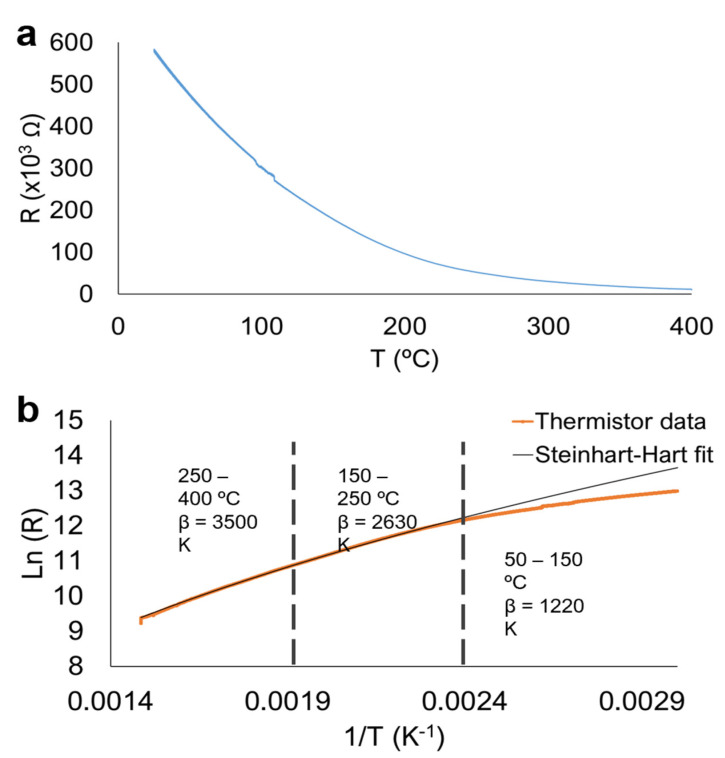
Resistance variation of the diamond NTC thermistor in the range 25–400 °C. (**a**) Measured curve. (**b**) Linear regression and fitting according to the “beta” and Steinhart-Hart models.

**Figure 4 materials-14-07334-f004:**
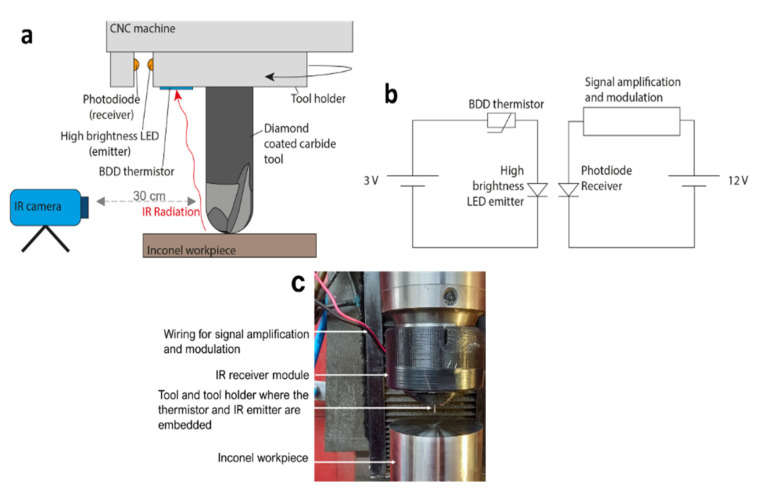
Temperature monitoring during face milling of the Inconel 718 workpiece. (**a**) Overall setup for simultaneous temperature monitoring with the IR camera and BDD thermistor. (**b**) Wireless LED communication circuit. (**c**) Image showing the real positioning of the machining/sensing elements described in (**a**).

**Figure 5 materials-14-07334-f005:**
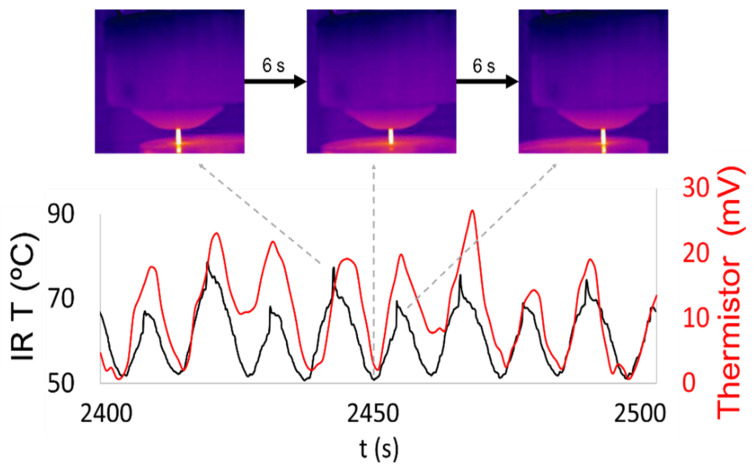
Temperature measurement recorded by the thermographic camera showing occurrence of lower temperature at the tool tip when approximately at the middle of the cutting path and higher temperature at inflection positions. This curve was used for calibration of the diamond-based thermistor, for which the resistance variation induced a 0.84 mV/°C variation at the photodiode.

**Table 1 materials-14-07334-t001:** Parameters used for diamond growth by HFCVD.

Type of Coating	Layer	H_2_ (mL/min)	CH_4_ (mL/min)	Ar (mL/min)	T_filament_ (°C)	T_substrate_ (°C)	Pressure (kPa)
Undoped diamond	SMCD	200	4	0	2300	850	15
	NCD	200	8	0	2250	800	10
Boron doped diamond	MCD	100	4	5	2300	700	75

## Data Availability

The data presented in this study are available on request from the corresponding author.

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
