# Peer review of "Boron Doped Diamond for Real-Time Wireless Cutting Temperature Monitoring of Diamond Coated Carbide Tools"

_materials, 2021, doi:10.3390/ma14237334_

Round 1

Reviewer 1 Report

This paper presents boron doped diamond for real-time wireless cutting temperature monitoring of diamond coated carbide tools . The study, diamond/WC-Co substrates were coated with BDD by Hot Filament Chemical Vapor Deposition. Paper shows scanning electron microscopy, raman spectroscopy and van der Pauw method  used for morphological, structural, and electrical characterization samples.

The article is well written, can be accepted in present form.

The manuscript could be accepted after minor modification.

Author Response

The authors are grateful for the reviewer’s comments.

Reviewer 2 Report

The work presents the possibility of using thin diamond layers to monitor the cutting temperature. The interesting work presents new possibilities of using diamond layers as a thermistors, which is directly related to the properties of the material itself.

Please, specify the mean size of the crystallite on the surface of the sample (SEM).

It is not only the boron-doped diamond layers that conduct electricity. In the future, an interesting work was to compare the properties of the doped and non-doped layers.

The work meets the requirements for publication, please supplement it.

Author Response

The authors are thankful for the reviewer’s comments. The mean crystallite size on the surface of the sample was included in the manuscript, in line 189.

Reviewer 3 Report

This article is devoted to the analysis of the possibility of using thermal sensors based on boron-doped diamond in temperature monitoring systems of metal-cutting tools. The work is of a deep and comprehensive nature and focuses on the manufacturing technology of this type of sensors and their temperature characteristics. In this part, there are no complaints about the results presented and the approaches used.
The technical part concerning the implementation of a wireless temperature monitoring system using a diamond sensor is rather poorly presented. The electronic circuit used by the authors Figure 4b is too elementary and very nonlinear. The data obtained with its help on the sensitivity of the sensor are purely indicator information demonstrating the principle operability of the proposed method.
There can be no question of any metrological nature of this kind of wireless data transmission about the temperature of the controlled object. The development of a real temperature monitoring system was not part of the authors' tasks. therefore, this approach can be considered acceptable.
In cases of continuing research in this direction, the authors need to study in detail alternative types of serial high-temperature thermistors made of boron nitride or silicon carbide, and the results of testing thermistors made of boron-doped monocrystalline synthetic diamond. For example , publications
V.S.Bormashov, S.G.Buga, V.D.Blank, M.S.Kuznetsov, S.A.Nosukhin, S.A.Terentyev, E. G.Pel' High-speed thermistors made of synthetic single-crystal diamonds, Instruments and experimental methods. 52 (2009) 738-742. doi: 10.1134/S0020441209050182.
V.D.Blank, S.G. Buga, V.S.Bormashov, S.A.Terentyev, M.S.Kuznetsov, S.A.Nosukhin, E. G. Pel, Pulse thermometers based on synthetic single-crystal diamonds doped with boron, Diamond and related materials. 16 (2007) 970-973. doi: 10.1016/J.diamond.2006.12.049.
I believe that the article can be published in a Special issue "CVD deposition and characterization of multilayer and thin films"

Author Response

The authors are thankful for the reviewer’s comments. Indeed, the electronic circuitry was not central to the present manuscript. The suggested publications have been considered and included in the manuscript as relevant works towards future improvement of the results presented in the manuscript.

Reviewer 4 Report

l. 55 "monitorization" monitoring

l. 57 "developed a multisensory" - "multisensory" is an adjective, i.e. there is a noun missing

The boron concentration of the boron doped diamond should be measured and reported.

Fig. 3, figure caption: "Resistance variation of the diamond NTC thermistor in the for temperatures between 25 and 400 ºC." "in the for"?

l. 241: "dopping level" doping

Author Response

The authors appreciate the reviewer’s input towards improving the English of the manuscript. The suggested modifications have been included in the manuscript.   

The doping concentration is indicated in lines 123 to 125 of the manuscript.